# PINF: Continuous Normalizing Flows for Physics-Constrained Deep Learning

## Abstract

The normalization constraint on probability density poses a significant challenge for solving the Fokker-Planck equation. Normalizing Flow, an invertible generative model leverages the change of variables formula to ensure probability density conservation and enable the learning of complex data distributions. In this paper, we introduce Physics-Informed Normalizing Flows (PINF), a novel extension of continuous normalizing flows, incorporating diffusion through the method of characteristics. Our method, which is mesh-free and causality-free, can efficiently solve high dimensional time-dependent and steady-state Fokker-Planck equations.

## 1 Introduction

The Fokker-Planck (FP) equation (Risken, 1996) is a well-known partial differential equation that describes the evolution of a stochastic system's probability density function (PDF) over time. Due to the high-dimensional variable and unbounded domain, traditional numerical methods, such as the finite difference methods (FDM) (Kumar & Narayanan, 2006), the finite element methods (FEM) (Deng, 2009), and the path integral methods (Wehner & Wolfer, 1983) prove to be computationally daunting in tackling the FP equations. By contrast, deep learning algorithms (Sirignano & Spiliopoulos, 2018; Raissi et al., 2019; Weinan & Yu, 2018), without a specific network structure, violate the normalization constraint on PDF, resulting in reduced accuracy and prevalent errors.

Recently, researchers have recognized the potential of flow-based generative models in learning complicated probability distributions (Kingma & Dhariwal, 2018; Ho et al., 2019; Albergo et al., 2019), alongside the connection to optimal transport (OT) theory (Finlay et al., 2020; Yang & Karniadakis, 2020; Zhang et al., 2018). Tang et al. (2022) employed normalizing flows as alternative solutions and proposed the KRnet for solving the steady-state Fokker-Planck (SFP) equations. However, the discrete normalizing flow remains circumscribed to model a single target distribution at a time, typically the final or steady-state distribution, thus constraining its utility in the time-dependent Fokker-Planck (TFP) equations. Consequently, Feng et al. (2022) introduced the temporal normalizing flow (TNF) to estimate time-dependent distributions for TFP equations, albeit without encapsulating the inherent physical laws of FP equations in structure.

More naturally, we generalize the continuous normalizing flow (CNF) (Chen et al., 2018) with diffusion and propose a novel intelligent architecture: Physics-Informed Normalizing Flows (PINF) for solving FP equations. We encode the physical constraints into ordinary differential equations (ODEs) using the method of characteristics and train the model in a self-supervised way. Numerical experiments demonstrate both accuracy and efficiency in solving high-dimensional TFP and SFP equations, even without the need for meticulous hyperparameter tuning.

## 2 Problem Definition

Here we first give a brief introduction to the FP equation. Consider the state variable $\mathbf{X}_t \in \mathbb{R}^d$ described by the stochastic differential equation (SDE) (Oksendal, 2013):

$$d\mathbf{X}_t = \mu(\mathbf{X}_t, t)dt + \sigma(\mathbf{X}_t, t)d\boldsymbol{W}_t, \tag{1}$$

where the drift coefficient $\mu(\mathbf{X}_t, t) \in \mathbb{R}^d$ is a vector field, $\sigma(\mathbf{X}_t, t) \in \mathbb{R}^{d \times M}$ is a matrix-valued function and $\boldsymbol{W}_t$ is an $M$-dimensional standard Wiener process. The probability density function

$p(\boldsymbol{x}, t)$ of $\mathbf{X}_t$ satisfies the following Fokker-Planck equation:

$$\frac{\partial p(\boldsymbol{x}, t)}{\partial t} = -\nabla \cdot [p(\boldsymbol{x}, t)\mu(\boldsymbol{x}, t)] + \nabla \cdot [\nabla \cdot (p(\boldsymbol{x}, t)\boldsymbol{D}(\boldsymbol{x}, t))], \qquad \forall (\boldsymbol{x}, t) \in \mathbb{R}^d \times \mathbb{R}^+,$$

$$p(\boldsymbol{x}, 0) = p_0(\boldsymbol{x}),$$

$$p(\boldsymbol{x}) \to 0 \quad \text{as} \quad ||\boldsymbol{x}|| \to \infty,$$

$\qquad(2)$

where $(\boldsymbol{x}, t) \in \mathbb{R}^{d+1}$ denote the spatial-temporal variables, $\boldsymbol{D}(\boldsymbol{x}, t) = \frac{1}{2}\sigma(\boldsymbol{x}, t)\sigma(\boldsymbol{x}, t)^T$ is the diffusion matrix, $p_0(\boldsymbol{x})$ is the initial PDF of $\mathbf{X}_t$, $p(\boldsymbol{x}, t)$ is defined with the unbounded boundary condition, $\nabla$ is an operator for spatial variables and $||\boldsymbol{x}||$ indicates the $\ell_2$ norm of $\boldsymbol{x}$.

The stationary solution of Eq.(2) means the invariant measure independent of time, satisfying

$$-\nabla \cdot [p(\boldsymbol{x}, t)\mu(\boldsymbol{x}, t)] + \nabla \cdot [\nabla \cdot (p(\boldsymbol{x}, t)\boldsymbol{D}(\boldsymbol{x}, t))] = 0, \qquad \forall (\boldsymbol{x}, t) \in \mathbb{R}^d \times \mathbb{R}^+ \qquad (3)$$

More specifically, i.e.,

$$-\sum_{i=1}^{d} \frac{\partial}{\partial x_i} [p(\boldsymbol{x}, t)\mu_i(\boldsymbol{x}, t)] + \sum_{i=1}^{d}\sum_{j=1}^{d} \frac{\partial^2}{\partial x_i \partial x_j} [p(\boldsymbol{x}, t)D_{ij}(\boldsymbol{x}, t)] = 0, \qquad \forall (\boldsymbol{x}, t) \in \mathbb{R}^d \times \mathbb{R}^+ \quad (4)$$

For the physical background of the FP equation, there are some extra constraints on its solution $p(\boldsymbol{x}, t)$:

$$\int_{\mathbb{R}^d} p(\boldsymbol{x}, t)d\boldsymbol{x} = 1, \qquad \forall t \in \mathbb{R}^+$$

$$p(\boldsymbol{x}, t) \geq 0, \qquad \forall (\boldsymbol{x}, t) \in \mathbb{R}^d \times \mathbb{R}^+,$$

$\qquad(5)$

which are called normalization and nonnegativity constraints.

## 3 RELATED WORK

To better describe our algorithm and its connections with flow-based models, it is worthwhile to review prior research on normalizing flows.

### 3.1 NORMALIZING FLOWS AND CHANGE OF VARIABLES

Given a latent variable $\boldsymbol{z} \in \mathbb{Z} \subset \mathbb{R}^d$ drawn from a simple prior probability distribution $p_Z$. When normalizing flows transform $\boldsymbol{z}$ into $\boldsymbol{x} = f(\boldsymbol{z}) \in \mathbb{X} \subset \mathbb{R}^d$ using a bijection $f : \mathbb{Z} \to \mathbb{X}$, the probability density function of $\boldsymbol{x}$ follows the change of variables formula:

$$p_Z(\boldsymbol{z}) = p_X(\boldsymbol{x}) \left| \det\left( \frac{\partial f(\boldsymbol{z})}{\partial \boldsymbol{z}^T} \right) \right|, \qquad (6)$$

where $\frac{\partial f(\boldsymbol{z})}{\partial \boldsymbol{z}^T}$ is the Jacobian of $f$ at $\boldsymbol{z}$.

To describe a complex target distribution, a sequence of invertible and learnable mappings represented as $f_\theta = f_K \circ \cdots \circ f_2 \circ f_1$ are constructed and $\theta$ denotes the trainable parameter of the neural network. Let $\boldsymbol{z}_1 = \boldsymbol{z}, \boldsymbol{z}_{k+1} = f_k(\boldsymbol{z}_k)(k = 1, \cdots, K)$, then the log density of $\boldsymbol{x} = f_\theta(\boldsymbol{z})$ is computed by

$$\log p_X(\boldsymbol{x}) = \log p_Z(\boldsymbol{z}) - \sum_{k=1}^{K} \log \left| \det\left( \frac{\partial f_k(\boldsymbol{z}_k)}{\partial \boldsymbol{z}_k^T} \right) \right|. \qquad (7)$$

There are some simple optional structures, such as planar flows and radial flows (Rezende & Mohamed, 2015). A key challenge is to increase the representative power of normalizing flows while simplifying the computation of the associated Jacobian determinant. NICE (Dinh et al., 2015) employed affine coupling layers, duplicating a portion of the input while transforming the remaining part, thereby maintaining reversibility. Real NVP (Dinh et al., 2016) introduced scale and translation parameters, further improving the performance of the flow. Additionally, the Jacobian of Real NVP transformations has a specific lower triangular structure, ensuring efficient computations with linear time complexity $\mathcal{O}(d)$ for the logdet-Jacobian term.

It is important to note that the resulting target distribution adheres to the probability normalization constraint, as guaranteed by the change of variables formula.

$$1 = \int_{\Omega_\mathbb{Z}} p_Z(\boldsymbol{z})d\boldsymbol{z} = \int p_X(\boldsymbol{x}) \left| \det\left( \frac{\partial f_\theta(\boldsymbol{z})}{\partial \boldsymbol{z}^T} \right) \right| d\boldsymbol{z} = \int_{\Omega_\mathbb{X}} p_X(\boldsymbol{x})d\boldsymbol{x} \qquad (8)$$

### 3.2 Continuous normalizing flows and instantaneous change of variables

Chen et al. (2018) proposed the Neural ODE framework and derived the finite normalizing flows to a continuous limit scheme, effectively expressing the invertible mappings using ODEs. The latent variable $z$ and its log probability change according to the instantaneous change of variables theorem (Villani, 2003):

$$\begin{cases} \dfrac{d\boldsymbol{z}(t)}{dt} = f_\theta(\boldsymbol{z}, t) \\ \dfrac{d\log p\left(\boldsymbol{z}(t), t\right)}{dt} = -tr\left(\dfrac{\partial f_\theta}{\partial z}\right) = -\nabla \cdot f_\theta \end{cases} \tag{9}$$

By leveraging the adjoint sensitivity method to resolve augmented ODEs in reverse time, CNF computes gradients with respect to $\theta$ and is trained directly using maximum likelihood. An unexpected side-benefit is that the likelihood can be calculated using relatively cheap trace operations instead of the Jacobian determinant. Moreover, the entire transformation is automatically bijective when Eq.(9) has a unique solution. Therefore, the mapping $f_\theta$ does not need to be bijective.

### 3.3 Relationship between continuous normalizing flows and Fokker-Planck equations

Continuous normalizing flows can be related to the special case of the FP equation with zero diffusion, known as the Liouville equation. Let $\boldsymbol{z}(t) \in \mathbb{R}^d$ evolve through time according to the degenerate Eq.(1): $\frac{d\boldsymbol{z}(t)}{dt} = f(\boldsymbol{z}(t), t)$. As shown in Eq.(2), the probability density function $p(\boldsymbol{z}, t)$ satisfies the FP equation represented as:

$$\frac{\partial p(\boldsymbol{z}, t)}{\partial t} = -\nabla \cdot [p(\boldsymbol{z}, t) f(\boldsymbol{z}, t)]. \tag{10}$$

To compute the value of $p(\boldsymbol{z}, t)$, the characteristics method tracks the trajectory of a particle $\boldsymbol{z}(t)$. The total derivative of $p(\boldsymbol{z}(t), t)$ is given by

$$\frac{dp(\boldsymbol{z}(t), t)}{dt} = \frac{\partial p(\boldsymbol{z}, t)}{\partial t} + \frac{\partial p(\boldsymbol{z}, t)}{\partial \boldsymbol{z}} \cdot \frac{d\boldsymbol{z}}{dt} = -\nabla \cdot (pf) + \nabla p \cdot f = -p(\nabla \cdot f). \tag{11}$$

By dividing $p$ on both sides, we obtain the same result as Eq.(9). (See Appendix A.2 in Chen et al. (2018) for more details).

## 4 PINF: Physics-Informed Normalizing Flows

Let us begin by introducing the PINF algorithm for time-dependent FP equations, categorizing them into two scenarios based on whether the diffusion term is zero. Subsequently, we will present the special design of the algorithm for solving the steady-state FP equation.

### 4.1 Time-dependent Fokker-Planck equations

The TFP equation is essentially an initial value problem (2), where the initial density function is known.

#### 4.1.1 TFP equation with zero diffusion

**Problem Setup:**

$$\frac{\partial p(\boldsymbol{x}, t)}{\partial t} = -\nabla \cdot [p(\boldsymbol{x}, t) \mu(\boldsymbol{x}, t)] \tag{12}$$
$$p(\boldsymbol{x}, 0) = p_0(\boldsymbol{x})$$

The objective of solving the TFP equation is to compute the density $p$ at any given point $(\boldsymbol{x}', t')$. Following the derivation in Eq.(11), the TFP equation can be reformulated as an initial value problem of ODEs.

$$\begin{cases} \dfrac{d\boldsymbol{x}(t)}{dt} = \mu(\boldsymbol{x}, t), \quad \boldsymbol{x}(t') = \boldsymbol{x}' \\ \dfrac{d\log p\left(\boldsymbol{x}(t), t\right)}{dt} = -\nabla \cdot \mu(\boldsymbol{x}, t) \end{cases} \tag{13}$$

Notably, the increment in $\log p$ is independent of its value and solely depends on the start time $t'$, the stop time $t_0 = 0$, and the value of $\boldsymbol{x}$. Therefore, the initial states for $(\boldsymbol{x}, \log p)$ can be set as $(\boldsymbol{x}', 0)$, and the solution is obtained by using the corresponding output of ODE solvers, yielding $(\boldsymbol{x}_0, \Delta \log p)$. The specific algorithm is outlined as follows:

---

**Algorithm 1** PINF algorithm for TFP equations with zero diffusion

---

**Input:** drift term $\mu(\boldsymbol{x}, t)$, samples $\boldsymbol{x}'$, time $t'$, initial PDF $p_0(\boldsymbol{x})$.
    **def** $f_{aug}([\boldsymbol{x}_t, \log p_t], t)$:                             ▷ ODEs dynamics
        **return** $[\mu, -\nabla \cdot \mu]$             ▷ Concatenate dynamics of state and log-density
    $[\boldsymbol{x}_0, \Delta \log p] \leftarrow \text{ODESolve}(f_{aug}, [\boldsymbol{x}', 0], t', 0)$     ▷ Calculate $\int_{t'}^{0} f_{aug}([\boldsymbol{x}(t), \log p(\boldsymbol{x}(t), t)], t) \, dt$
    $\log \hat{p} \leftarrow \log p_0(\boldsymbol{x}_0) - \Delta \log p$                          ▷ Add change in log-density
    $\hat{p}(\boldsymbol{x}', t') = e^{\log \hat{p}}$
**Output:** $\hat{p}(\boldsymbol{x}', t')$

---

The key distinction between PINF and CNF lies in the fact that the drift $\mu(\boldsymbol{x}, t)$ is known in the TFP equation. In this context, there is no need to train $f_\theta$ from the real data samples and we can simply use ODE solvers once to obtain the outcomes.

### 4.1.2 TFP EQUATION WITH DIFFUSION

**Problem Setup:**

$$\frac{\partial p(\boldsymbol{x}, t)}{\partial t} = -\nabla \cdot [p(\boldsymbol{x}, t)\mu(\boldsymbol{x}, t)] + \nabla \cdot [\nabla \cdot (p(\boldsymbol{x}, t)\boldsymbol{D}(\boldsymbol{x}, t))]$$
$$p(\boldsymbol{x}, 0) = p_0(\boldsymbol{x}) \tag{14}$$

We first perform some necessary transformations of the equation.

$$\begin{aligned}
\frac{\partial p(\boldsymbol{x}, t)}{\partial t} &= -\nabla \cdot (p\mu) + \nabla \cdot [\nabla \cdot (p\boldsymbol{D})] \\
&= -\nabla \cdot [p\mu - \nabla \cdot (p\boldsymbol{D})] \\
&= -\nabla \cdot [p\mu - (\nabla p)\boldsymbol{D} - p(\nabla \cdot \boldsymbol{D})] \\
&= -\nabla \cdot [p(\mu - (\nabla \log p)\boldsymbol{D} - \nabla \cdot \boldsymbol{D})] \\
&= -\nabla \cdot (p\mu^*)
\end{aligned} \tag{15}$$

Let us define $\mu^*(\boldsymbol{x}, t) := \mu - (\nabla \log p)\boldsymbol{D} - \nabla \cdot \boldsymbol{D}$. We force $\boldsymbol{x}$ to evolve through time following $\frac{d\boldsymbol{x}}{dt} = \mu^*(\boldsymbol{x}, t)$ and get the associated ODEs by Eq.(11).

$$\begin{cases}
\dfrac{d\boldsymbol{x}(t)}{dt} = \mu^*(\boldsymbol{x}, t) \\
\dfrac{d\log p(\boldsymbol{x}(t), t)}{dt} = -\nabla \cdot \mu^*(\boldsymbol{x}, t)
\end{cases} \tag{16}$$

In particular, unlike the zero diffusion case, the ODE dynamics depend on the unknown function $\log p$ and its gradient will not be evaluated during the forward calculation. For this reason, we parameterize $\log p(\boldsymbol{x}, t)$ as a neural network $\phi_\theta$.

**Neural network architecture.** The network structure is inspired by the value function representations used for stochastic optimal control (Li et al., 2022), multi-agent optimal control (Onken et al., 2021b), and high-dimensional optimal control (Onken et al., 2022). The network is given by

$$u(\boldsymbol{s}; \theta) = \boldsymbol{w}^\top \mathcal{N}(\boldsymbol{s}; \theta_\mathcal{N}) + \frac{1}{2}\boldsymbol{s}^\top(\boldsymbol{A}^\top \boldsymbol{A})\boldsymbol{s} + \boldsymbol{b}^\top \boldsymbol{s} + c, \tag{17}$$

where $\theta = (\boldsymbol{w}, \theta_\mathcal{N}, \boldsymbol{A}, \boldsymbol{b}, c)$ are the trainable weights of $u_\theta$. The shapes of those parameters are listed as follows: the inputs $\boldsymbol{s} = (\boldsymbol{x}, t) \in \mathbb{R}^{d+1}$ corresponding to space–time, $\mathcal{N}(\boldsymbol{s}; \theta_\mathcal{N}) : \mathbb{R}^{d+1} \rightarrow \mathbb{R}^m$, $\boldsymbol{w} \in \mathbb{R}^m$, $\boldsymbol{A} \in \mathbb{R}^{r \times (d+1)}$, $\boldsymbol{b} \in \mathbb{R}^{d+1}$, and $c \in \mathbb{R}$. The rank $r = \min(10, d+1)$ is set to limit the number of parameters in $\boldsymbol{A}^\top \boldsymbol{A}$. Here, $\boldsymbol{A}$, $\boldsymbol{b}$, and $c$ model quadratic potentials, i.e., linear dynamics; $\mathcal{N}$ models nonlinear dynamics.

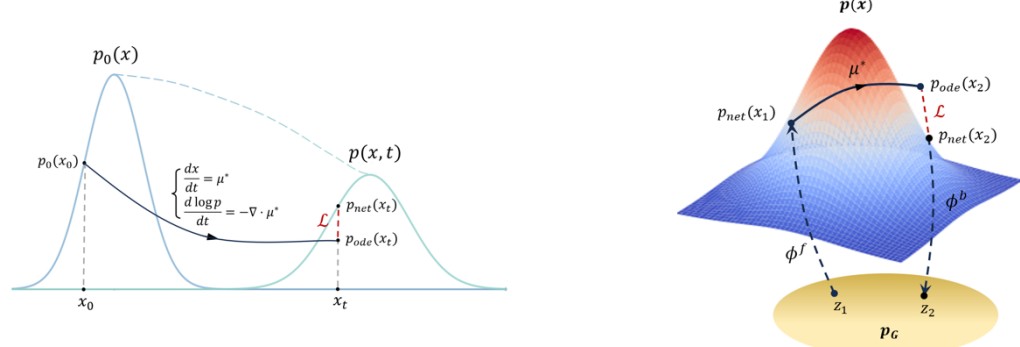

Figure 1: An illustration of the PINF algorithm for TFP equations (left) and SFP equations (right).

In our implementation, for $\mathcal{N}$, we use a residual neural network (ResNet) (He et al., 2016) with $(L+1)$ layers and obtain $\mathcal{N}(\boldsymbol{s}; \theta_\mathcal{N}) = \boldsymbol{a}_L$ as the final step of the forward propagation

$$
\begin{aligned}
\boldsymbol{a}_0 &= \sigma(\boldsymbol{K}_0 \boldsymbol{s} + \boldsymbol{b}_0), \\
\boldsymbol{a}_k &= \boldsymbol{a}_{k-1} + h\sigma(\boldsymbol{K}_k \boldsymbol{a}_{k-1} + \boldsymbol{b}_k), \quad (1 \le k \le L)
\end{aligned}
\tag{18}
$$

where the trainable weights are $\boldsymbol{K}_0 \in \mathbb{R}^{m \times (d+1)}, \boldsymbol{K}_k \in \mathbb{R}^{m \times m}(1 \le k \le L), \boldsymbol{b}_k \in \mathbb{R}^m (0 \le k \le L)$, and $\theta_\mathcal{N} = \{(K_k, b_k)\}_{k=0}^L$. We select the step size $h = 1/L$ and the element-wise activation function $\sigma(x) = \log(\exp(x) + \exp(-x))$ (Onken et al., 2021a), which is the antiderivative of the hyperbolic tangent, i.e., $\sigma'(x) = \tanh(x)$. It can also be seen as a smoothed absolute value function (Ruthotto et al., 2020).

To satisfy the initial value condition $p(\boldsymbol{x}, 0) = p_0(\boldsymbol{x})$, the network $\phi_\theta$ is cast as

$$
\phi_\theta(\boldsymbol{x}, t) = \log p_0(\boldsymbol{x}) + tu(\boldsymbol{x}, t; \theta),
\tag{19}
$$

to represent $\log p(\boldsymbol{x}, t)$ (Lagaris et al., 1998). This structured design can impose hard constraints that are strictly enforced, rather than soft constraints (Wang & Yu, 2021).

**Self-supervised training method.** Distinct from normalizing flows that minimize the Kullback-Leibler divergence, we employ a self-supervised training method. Our approach eliminates the need for labeled data or real data samples from the target distribution, which are sometimes challenging or costly to acquire. Training data is adaptively generated based on the initial PDF.

For training $\phi_\theta$, we solve the ODEs (16) related to the network $\phi_\theta$, obtaining predictions in the form of $\log p_{\text{ode}}$. Moreover, the network can also output the prediction $\log p_{\text{net}}$ directly. We calculate the Mean Squared Error (MSE) between $\log p_{\text{ode}}$ and $\log p_{\text{net}}$ and use the Adam optimizer (Kingma & Ba, 2014).

$$
\mathcal{L} = \frac{1}{N} \sum_{k=1}^N \| \log p_{\text{ode}}(\boldsymbol{x}_k, t_k) - \log p_{\text{net}}(\boldsymbol{x}_k, t_k) \|^2
\tag{20}
$$

To address issues such as small gradients and optimization difficulties in high-dimensional scenarios, we avoid comparing $p_{\text{pred}}$ directly in the loss function: $\mathcal{L} = \text{MSE}(p_{\text{ode}}, p_{\text{net}})$. Experimental results also demonstrate that applying the logarithm function accelerates the training process.

**Flexible prediction modes.** With the trained $\phi_\theta$, we provide two flexible prediction modes: numerical solutions solved by ODE solvers and continuous solutions predicted by neural networks. This flexibility enables a trade-off between efficiency and accuracy, maintaining advantages such as memory efficiency, adaptive solvers, and parallel computation (Kang et al., 2021). When using neural networks for prediction, our approach is mesh-free and causality-free (Nakamura-Zimmerer et al., 2020), similar to PINNs. Data at different time steps can be computed rapidly and in parallel, allowing for real-time or low-power applications. Alternatively, when using ODE solvers for prediction, we can freely choose from modern ODE solvers for adaptive computation (Runge, 1895; Wanner & Hairer, 1996). See Alg.(2) for the complete algorithm.

---

**Algorithm 2** PINF algorithm for TFP equations with diffusion

---

**Input:** drift term $\mu(\boldsymbol{x}, t)$, diffusion matrix $\boldsymbol{D}(\boldsymbol{x}, t)$, samples $\boldsymbol{x}$, time $t$, initial PDF $p_0(\boldsymbol{x})$, stop time $T$, maximum iteration number $M$, learning rate $\eta$.

    **def** $f_{aug}([\boldsymbol{x}_t, \log p_t], t)$:                                              ▷ ODEs dynamics

             $\mu^* \leftarrow \mu - (\nabla \phi_\theta)\boldsymbol{D} - \nabla \cdot \boldsymbol{D}$                 ▷ Characteristic curves $\frac{d\boldsymbol{x}(t)}{dt}$

             **return** $[\mu^*, -\nabla \cdot \mu^*]$           ▷ Concatenate dynamics of state and log-density

**Train** $\phi_\theta$:

**for** $k = 0, \cdots, M$:

         Uniformly sample $t_k \in [0, T]$                                    ▷ Sample training data

         Sample mini-batch $\boldsymbol{x}_0^k \sim p_0(\boldsymbol{x})$

         $[\boldsymbol{x}^k, \log p(\boldsymbol{x}^k, t_k)] \leftarrow \text{ODESolve}(f_{aug}, [\boldsymbol{x}_0^k, \log p_0(\boldsymbol{x}_0^k)], 0, t_k)$

                              ▷ Calculate $[\boldsymbol{x}_0^k, \log p_0(\boldsymbol{x}_0^k)] + \int_0^{t_k} f_{aug}([\boldsymbol{x}(t), \log p(\boldsymbol{x}(t), t)], t)\, dt$

         ODEs prediction: $\log p_{\text{ode}} \leftarrow \log p(\boldsymbol{x}^k, t_k)$             ▷ Two prediction modes

         $\phi_\theta$ prediction: $\log p_{\text{net}} \leftarrow \phi_\theta(\boldsymbol{x}^k, t_k)$

         Compute the MSE loss: $\mathcal{L} = \text{MSE}(\log p_{\text{ode}}, \log p_{\text{net}})$      ▷ Calculate loss on mini-batch $\boldsymbol{x}^k$

         Update the parameters $\theta$ using the Adam optimizer with learning rate $\eta$.     ▷ Train $\phi_\theta$ once

**Predict** $\hat{p}$:

$\phi_\theta$ mode: $\hat{p}_{\text{net}}(\boldsymbol{x}, t) = e^{\phi_\theta(\boldsymbol{x}, t)}$

ODEs mode: $[\boldsymbol{x}_0, \Delta \log p] \leftarrow \text{ODESolve}(f_{aug}, [\boldsymbol{x}, 0], t, 0)$

         $\log \hat{p}_{\text{ode}} \leftarrow \log p_0(\boldsymbol{x}_0) - \Delta \log p$                 ▷ Add change in log-density

         $\hat{p}_{\text{ode}}(\boldsymbol{x}, t) = e^{\log \hat{p}_{\text{ode}}}$

**Output:** $\hat{p}_{\text{net}}(\boldsymbol{x}, t), \hat{p}_{\text{ode}}(\boldsymbol{x}, t)$

---

## 4.2 STEADY-STATE FOKKER-PLANCK EQUATIONS

**Problem Setup:**

$$\frac{\partial p(\boldsymbol{x}, t)}{\partial t} = -\nabla \cdot [p(\boldsymbol{x}, t)\mu(\boldsymbol{x}, t)] + \nabla \cdot [\nabla \cdot (p(\boldsymbol{x}, t)\boldsymbol{D}(\boldsymbol{x}, t))]$$
$$\frac{\partial p(\boldsymbol{x}, t)}{\partial t} = 0 \tag{21}$$

The SFP equation replaces the initial value condition with an equation constraint (3), ensuring that its solution remains invariant over time. Therefore, the steady-state PDF only depends on spatial variables $\boldsymbol{x}$ and we derive another form of our problem:

$$-\nabla \cdot [p(\boldsymbol{x})\mu] + \nabla \cdot [\nabla \cdot (p(\boldsymbol{x})\boldsymbol{D}] = 0 \tag{22}$$

**Normalization Challenge.** In the TFP equations, since the initial PDF $p_0(\boldsymbol{x})$ satisfies the normalization constraint, the total density integral on the space domain $\int_{\mathbb{R}^d} p(\boldsymbol{x}, t)d\boldsymbol{x}$ remains conserved as the solution evolves under equation constraints or ODEs controls. Consequently, the solution $p(\boldsymbol{x}, t)$ predicted by the PINF algorithm also satisfies the normalization constraint.

However, for the SFP equations, it is easily checked that if $p(\boldsymbol{x})$ is the solution, then any $cp(\boldsymbol{x})$ ($c > 0$ is a constant) also satisfies the equation. In the ODEs, the scheme remains unchanged due to $\nabla \log(cp(\boldsymbol{x})) = \nabla(\log p(\boldsymbol{x}) + \log c) = \nabla \log p(\boldsymbol{x})$. Hence, starting from $\boldsymbol{x}_1$ and evolving through $t_1$ to $t_2$, both ODEs associated with $p(\boldsymbol{x})$ and $cp(\boldsymbol{x})$ reach the same point $\boldsymbol{x}_2$ and have the same increment $\Delta \log p$. For the ODEs of $cp(\boldsymbol{x})$, the predicted value of $\log \hat{p}(\boldsymbol{x}_2)$ is given by

$$\begin{aligned}
\log \hat{p}(\boldsymbol{x}_2) &= \log(cp(\boldsymbol{x}_1)) + \Delta \log p \\
&= \log c + (\log p(\boldsymbol{x}_1) + \Delta \log p) \\
&= \log c + \log p(\boldsymbol{x}_2) \\
&= \log(cp(\boldsymbol{x}_2))
\end{aligned} \tag{23}$$

Ultimately, $p(\boldsymbol{x})$ and $cp(\boldsymbol{x})$ satisfy the ODEs.

**Neural network architecture.** We still employ a neural network $\phi_\theta(\boldsymbol{x})$ to parameterize $\log p(\boldsymbol{x})$ in ODEs (16). For the normalization constraint, we adopt the Real NVP (Dinh et al., 2016) to model the equilibrium distribution from a Gaussian distribution $p_G(\boldsymbol{z}; \boldsymbol{0}, \boldsymbol{I})$. We stack $L$ affine coupling

layers to build a flexible and tractable bijective transformation. At each layer, given the input $\boldsymbol{x} \in \mathbb{R}^d$ and dimension $n < d$, the output $\boldsymbol{y}$ is defined as

$$
\begin{aligned}
\boldsymbol{y}_{1:n} &= \boldsymbol{x}_{1:n} \\
\boldsymbol{y}_{n+1:d} &= \boldsymbol{x}_{n+1:d} \odot \exp\left(s(\boldsymbol{x}_{1:n})\right) + t(\boldsymbol{x}_{1:n})
\end{aligned}
\tag{24}
$$

where $s$ and $t$ represent scale and translation, and $\odot$ is the element-wise product. The Jacobian of this transformation is triangular, reads

$$
\frac{\partial \boldsymbol{y}}{\partial \boldsymbol{x}^T} = \left[ \begin{array}{cc} \mathbf{1}_{n \times n} & 0 \\ \frac{\partial \boldsymbol{y}_{n+1:d}}{\partial \boldsymbol{x}_{1:n}^T} & \mathrm{diag}\left(\exp\left[s(\boldsymbol{x}_{1:n})\right]\right) \end{array} \right]
\tag{25}
$$

Consequently, we can efficiently compute its log-determinant as $\sum_k s(\boldsymbol{x}_{1:n})_k$.

Since partitioning can be implemented using a binary mask $b$, the forward and backward computation processes respectively follow the equations,

$$
\begin{aligned}
\text{Forward:} \quad \boldsymbol{x}_b &= b \odot \boldsymbol{x} \\
\boldsymbol{y} &= \boldsymbol{x}_b + (1-b) \odot \left(\boldsymbol{x} \odot \exp\left(s(\boldsymbol{x}_b)\right) + t(\boldsymbol{x}_b)\right) \\
\log\left|\det\left(\frac{\partial \boldsymbol{y}}{\partial \boldsymbol{x}^T}\right)\right| &= \sum (1-b) \odot s(\boldsymbol{x}_b) \\
\text{Backward:} \quad \boldsymbol{y}_b &= b \odot \boldsymbol{y} \\
\boldsymbol{x} &= \boldsymbol{y}_b + (1-b) \odot \left(\boldsymbol{y} - t(\boldsymbol{y}_b)\right) \odot \exp\left(-s(\boldsymbol{y}_b)\right) \\
\log\left|\det\left(\frac{\partial \boldsymbol{x}}{\partial \boldsymbol{y}^T}\right)\right| &= -\sum (1-b) \odot s(\boldsymbol{y}_b)
\end{aligned}
\tag{26}
$$

Here, $s, t : \mathbb{R}^d \to \mathbb{R}^d$ are constructed via a simple 3-layer MLP (Goodfellow et al., 2016) with the activation function $\sigma(x) = \tanh(x)$.

$$
\begin{aligned}
\boldsymbol{a}_1 &= \sigma(\boldsymbol{K}_1 \boldsymbol{x} + \boldsymbol{b}_1), \\
\boldsymbol{a}_2 &= \sigma(\boldsymbol{K}_2 \boldsymbol{a}_1 + \boldsymbol{b}_2), \\
\boldsymbol{a}_3 &= \boldsymbol{K}_3 \boldsymbol{a}_2 + \boldsymbol{b}_3.
\end{aligned}
\tag{27}
$$

The self-supervised training method and loss function remain consistent with the previous description. This is the PINF algorithm for SFP equations.

---

**Algorithm 3** PINF algorithm for SFP equations

---

**Input:** drift term $\mu$, diffusion matrix $\boldsymbol{D}$, samples $\boldsymbol{x}$, maximum iteration number $M$, learning rate $\eta$.

  **def** $f_{aug}([\boldsymbol{x}_t, \log p_t], t)$:     ▷ ODEs dynamics

    $\mu^* \leftarrow \mu - (\nabla \phi_\theta) \boldsymbol{D} - \nabla \cdot \boldsymbol{D}$     ▷ Characteristic curves $\frac{d\boldsymbol{x}(t)}{dt}$

    **return** $[\mu^*, -\nabla \cdot \mu^*]$     ▷ Concatenate dynamics of state and log-density

  **Train** $\phi_\theta$:

  **for** $k = 0, \cdots, M$:

    Sample mini-batch $\boldsymbol{z}_0^k \sim p_G(\boldsymbol{z})$     ▷ Sample training data

    Forward computation: $\boldsymbol{x}_0^k, \log p(\boldsymbol{x}_0^k) = \phi_\theta^f(z_0^k, \log p_G(z_0^k))$

    $[\boldsymbol{x}_1^k, \log p(\boldsymbol{x}_1^k)] \leftarrow \mathrm{ODESolve}(f_{aug}, [\boldsymbol{x}_0^k, \log p(\boldsymbol{x}_0^k)], 0, 1)$

        ▷ Calculate $[\boldsymbol{x}_0^k, \log p(\boldsymbol{x}_0^k)] + \int_0^1 f_{aug}([\boldsymbol{x}(t), \log p(\boldsymbol{x}(t), t)], t) \, dt$

    ODEs prediction: $\log p_{\mathrm{ode}} \leftarrow \log p(\boldsymbol{x}_1^k)$

    $\phi_\theta$ prediction: $\log p_{\mathrm{net}} \leftarrow \phi_\theta^b(\boldsymbol{x}_1^k)$

    Compute the MSE loss: $\mathcal{L} = \mathrm{MSE}(\log p_{\mathrm{ode}}, \log p_{\mathrm{net}})$     ▷ Calculate loss on mini-batch $\boldsymbol{x}_1^k$

    Update the parameters $\theta$ using the Adam optimizer with learning rate $\eta$.     ▷ Train $\phi_\theta$ once

  **Predict** $\hat{p}$:

    $\hat{p}_{\mathrm{net}}(\boldsymbol{x}) = e^{\phi_\theta^b(\boldsymbol{x})}$

**Output:** $\hat{p}_{\mathrm{net}}(\boldsymbol{x})$

---

## 5 EXPERIMENT

In this section, we consider the following three numerical examples to test the performance of our PINF algorithm.

### 5.1 A TOY EXAMPLE

We first present a toy example, a TFP equation with zero diffusion. We employ it to illustrate that our PINF algorithm is the method of characteristics in mathematics.

$$\frac{\partial p}{\partial t} + 2t\nabla \cdot (p\,\mathbf{1}) = 0, \quad t \in \mathbb{R}^+,\ \boldsymbol{x} \in \mathbb{R}^d$$
$$p(\boldsymbol{x}, 0) = (2\pi)^{-d/2}\exp\left(-\frac{1}{2}\|\boldsymbol{x}+\mathbf{1}\|^2\right) \tag{28}$$

Here $\mathbf{1} \in \mathbb{R}^d$ denotes an all-ones vector, and the diffusion term $\mu = 2t \cdot \mathbf{1}$ is independent of $\boldsymbol{x}$. The corresponding ODEs obtained through the PINF algorithm (1) are as follows:

$$\begin{cases} \dfrac{d\boldsymbol{x}(t)}{dt} = 2t \cdot \mathbf{1} \\ \dfrac{d\log p(\boldsymbol{x}(t),t)}{dt} = 0 \end{cases} \tag{29}$$

Consequently, the analytical solution of the equation is

$$p(\boldsymbol{x}, t) = p(\boldsymbol{x}_0, 0) = p_0(\boldsymbol{x} - t^2 \cdot \mathbf{1}) = (2\pi)^{-d/2}\exp\left(-\frac{1}{2}\|\boldsymbol{x} + (1-t^2)\mathbf{1}\|^2\right) \tag{30}$$

### 5.2 TFP EQUATION

We consider a relatively high-dimensional TFP equation. For this problem, the PINN is effective primarily with the dimension $d \leq 3$.

$$\frac{\partial p}{\partial t} - \frac{1}{2}\Delta p + 2\nabla \cdot (p\,\mathbf{1}) = 0, \quad t \in [0,1],\ \boldsymbol{x} \in \mathbb{R}^d$$
$$p(\boldsymbol{x}, 0) = (2\pi)^{-d/2}\exp\left(\|\boldsymbol{x}\|^2/2\right) \tag{31}$$

The exact solution is given by

$$p(\boldsymbol{x}, t) = \frac{1}{(2\pi(t+1))^{d/2}}\exp\left(-\frac{\|\boldsymbol{x} - 2t \cdot \mathbf{1}\|^2}{2(t+1)}\right) \tag{32}$$

Using the PINF algorithm (2), we formulate the ODEs as

$$\begin{cases} \dfrac{d\boldsymbol{x}(t)}{dt} = 2 \cdot \mathbf{1} - \dfrac{1}{2}\nabla \log p \\ \dfrac{d\log p(\boldsymbol{x}(t),t)}{dt} = -\nabla \cdot \left(\dfrac{d\boldsymbol{x}}{dt}\right) \end{cases} \tag{33}$$

We solve this TFP equation of $d = 10$. We take $L = 4$ residual layers with $m = 32$ hidden neurons and set training iteration $M = 10000$, the learning rate for Adam optimizer $\eta = 0.01$, and the batch size is 2000. The initial spatial training set is generated from $p_0(\boldsymbol{x})$, and the corresponding temporal training set is uniformly sampled in the interval $[0, T]$. We evaluate the performance of our PINF algorithm at the point $(\boldsymbol{x}_1, \boldsymbol{x}_2, 2, \ldots, 2) \in \mathbb{R}^d$ and the stop time $T = 1$, where $(\boldsymbol{x}_1, \boldsymbol{x}_2)$ is drawn from a uniform grid within the finite spatial domain $[-5, 5]^2$. The comparison between the prediction and the ground truth is shown in Fig.(2) and we observe a good agreement.

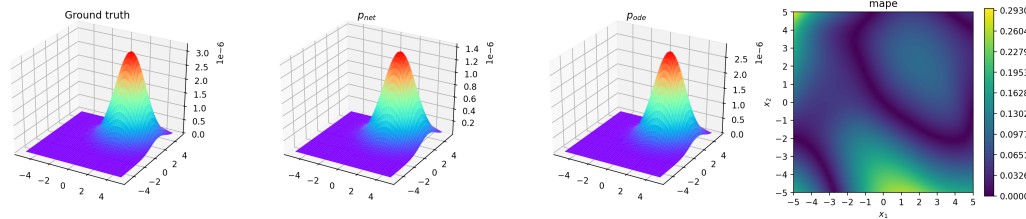

Figure 2: Exact and predicted solutions for the TFP equation from Case 2 at the stop time $T = 1$. Right panel: MAPE between the exact solution and the prediction $p_{\text{ode}}$.

### 5.3 HIGH DIMENSIONAL SFP EQUATION

In this part, a SFP equation with drift term $\mu(\boldsymbol{x}) = -a\boldsymbol{x}$ and diffusion matrix $\boldsymbol{D} = \frac{\sigma^2}{2}\mathbf{1}_{d\times d}$ is considered.

$$-\nabla \cdot (p(\boldsymbol{x})\mu) + \nabla \cdot [\nabla \cdot (p(\boldsymbol{x})\boldsymbol{D})] = 0, \quad \boldsymbol{x} \in \mathbb{R}^d \tag{34}$$

Its exact solution is a single Gaussian distribution, represented as

$$p(\boldsymbol{x}) = \left(\frac{a}{\pi\sigma^2}\right)^{d/2} \exp\left(-\frac{a\|\boldsymbol{x}\|^2}{\sigma^2}\right) \tag{35}$$

In our experiment, we set $a = \sigma = 1$ and evaluate our PINF algorithm on the SFP equation with dimensions $d = 30$ and $d = 50$. We employ $L = 4$ affine coupling layers and set $M = 500$, $\eta = 0.01$, batch size is 2000. Since its steady-state solution is an unimodal function near the origin, we select the test points as $(\boldsymbol{x}_1, \boldsymbol{x}_2, 0, \ldots, 0) \in \mathbb{R}^d$, where $(\boldsymbol{x}_1, \boldsymbol{x}_2)$ is drawn from a uniform grid within the range $[-3, 3]^2$, resulting in total $50 \times 50 = 2,500$ test points. Fig.(3) shows the exact solution $p(\boldsymbol{x})$ and our PINF solution $p(\boldsymbol{x}; \theta) = \phi_\theta^b(\boldsymbol{x})$, where it can be seen that they are visually indistinguishable, and the relative error remains below 0.2%, even for relatively high dimensions.

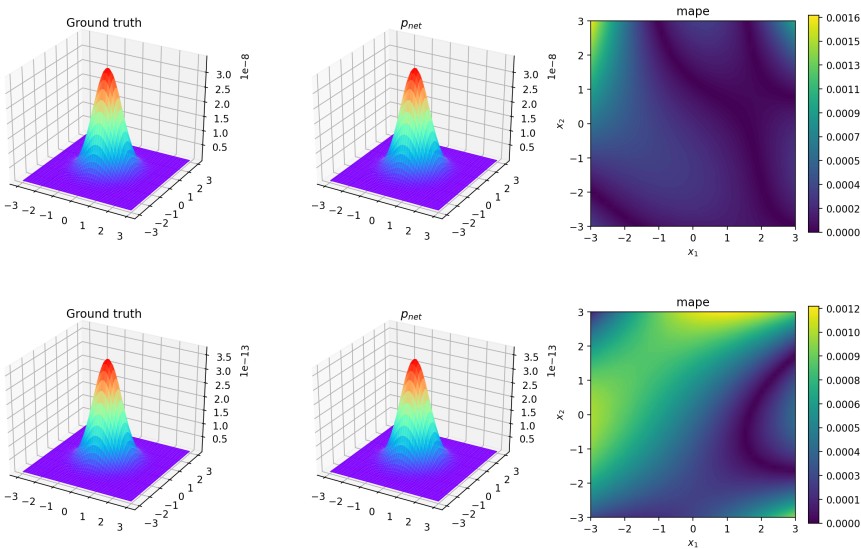

Figure 3: Exact and predicted solutions for the SFP equation from Case 3. Top row: d=30. Bottom row: d=50.

## 6 DISCUSSION

Our proposed PINF algorithm extends CNF to the scheme with diffusion using the method of characteristics. Compared to PINNs, which apply pointwise equation constraints, our improvements are mainly in reformulating the FP equation as ODEs and computing the integral of the loss along characteristic curves using an ODE solver, thereby enhancing training efficiency and stability. It effectively addresses the normalization constraint of the PDF even in high-dimensional FP problems.

The critical distinction between PINF for FP equations and CNF for density estimation lies in whether the drift term $\mu$ is known. In FP equations, the drift term $\mu(\boldsymbol{x}, t)$ is known and we obtain its solution by self-supervised training of a neural network. Conversely, the drift term $\mu_\theta$ necessitates likelihood-based training for the latter. In future work, we may explore the application of PINF with diffusion for density estimation tasks to enhance model generalization and data generation quality.

Considering the profound physical significance and wide-ranging applications of FP equations, the PINF algorithm can also be seamlessly integrated with large-scale mean-field games for optimal control problems, such as path planning, obstacle avoidance, and tracking in drone swarm applications.

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
