# OpenReview forum: "PINF: Continuous Normalizing Flows for Physics-Constrained Deep Learning"
_ICLR.cc/2024/Conference — ICLR 2024 Conference Withdrawn Submission_

### Official Review · Reviewer_QF8F · 2023-10-28

**Soundness:** 3 good
**Presentation:** 3 good
**Contribution:** 2 fair
**Rating:** 5
**Confidence:** 4

**Summary:**

The paper intends to solve Fokker-Planck equations, especially in high dimensional space. The method proposed here is to leverage a normalizing flow model to satisfy a hard constraint on distribution function (integral equals 1) and the training objective is self-supervised. In experiment, the method is tested on problems of dimension=30 and 50, and it shows the effectiveness of the method.

**Strengths:**

**Originality:** The paper has several novelties including combining normalizing flow to Fokker-Planck equations, designing a self-supervised loss calculated by the MSE loss between neural ODE prediction and network prediction.

**Quality:** The method is quite reasonable and exhibits some effectiveness in experiments.

**Clarity:** The method is presented clearly in the paper.

**Significance:** To applications of high-dimensional Fokker-Planck equations, this paper provides a useful tool.

**Weaknesses:**

The major weakness of the paper is insufficient experiment. For example, the authors could have compared their method with PINN with soft constraints and showed its advantage. Ablation study could be implemented on the combination of NF + SSL. What about NF + PINN loss? That is to use equation residual as loss.

**Questions:**

None.

---

### Official Review · Reviewer_RDE5 · 2023-10-30

**Soundness:** 2 fair
**Presentation:** 2 fair
**Contribution:** 3 good
**Rating:** 3
**Confidence:** 3

**Summary:**

The paper introduces a method to solve Fokker-Planck equations using continuous normalizing flows, with known drift and diffusion functions. While one can resolve the dynamics using an ordinary differential equation (ODE), it necessitates knowing the score of the distribution at a specific time, a challenging task. To address this, the authors employ a neural network to learn $\log p(x, t)$. The network is trained through a regression between i) $\log p$ obtained by solving the ODE and ii) the network's output. Presumably, an exact solution to this regression would mean precisely learning $\log p$. Additionally, the paper utilizes discrete normalizing flows, specifically RealNVP, to model an initial distribution that eventually diffuses into the steady-state solution of a Fokker-Planck equation. The proposed algorithms, both time-dependent and steady-state, have been validated on simple problems with known Gaussian distribution solutions, and the results showed good agreement.

Overall, the learning method is novel and interesting, but the paper is lacking in clarity, rigor and convincing, non-trivial experiments. I do not feel the paper meets the standard for ICLR in its current form, but could meet the standard at ICLR or another venue after these points have been addressed.

**Strengths:**

- Framing the problem as regression is clever – reminiscent of score matching, the technique behind diffusion models. Thus the central contribution of the paper is novel and interesting
- Most of the exposition of the background and theory is good, see weaknesses below for examples where this is not true
- The experiments shown seem to have worked very well. However, they are too easy (see below)

**Weaknesses:**

- Quite hard to understand some key points on a first reading. I only understood what was going on after looking at the algorithms. For example, it would have been very helpful to mention explicitly that by $\log p_{net}$ you mean $\phi$ (or just write $\phi$ in the equation instead of $\log p_{net}$)
- Doesn’t mention that the divergence of a network (trace of gradient) is only cheap (constant wrt data dimension) if you are happy with an approximation via the Hutchinson trace estimator. Otherwise the cost of the trace is the same as calculating the full Jacobian and scales linearly with data dimension. Seems to be the case in this work, but scaling behavior is not mentioned
- No justification in the text of why the specific parameterization of $\phi$ was chosen. It would be nice to get a rough picture without having to read the references. In addition, it is claimed that the design imposes hard constraints, but those constraints are not specified, nor is it clear why they are desirable
- There should be a proof that solving the regression problem is equivalent to $\phi = \log p$. It is clear that if $\phi = \log p$ then the MSE loss is zero. Is the reverse also true? It is implicitly assumed, and an explicit demonstration would greatly improve the theoretical soundness of the paper
- There is no mention of diffusion models or score matching, even though this is quite similar to this work. In both cases, you want to learn a simple function of $p(x, t)$ (here the log, in score matching the gradient of the log) which allows you to solve the dynamics of the system. Diffusion models are based on Fokker-Planck equations (see [1] for example)
- The “method of characteristics” is not explained even though it is mentioned multiple times. At least a reference that explains it is a must
- The PINN model is not explained or referenced, even though it is mentioned as a competitor
- The experiments are disappointing: all problems are essentially toy problems with linear drift functions and known Gaussian solutions. Clearly $\log p$ is just quadratic, so the form you have chosen for $\phi$ easily fits this. In fact, the neural net part of $\phi$ should probably just be 0. It’s nice that the method works on these problems and keeps working for higher dimensions, but the problems need to be harder to be convincing. In the discussion you mention applications of Fokker-Planck equations such as in optimal control. Why don’t you try one of these applications? If that’s outside the scope of this work, you should have some problems where the drift and/or diffusion are nonlinear. You can solve the ground-truth dynamics with an SDE and approximate the agreement between the ground-truth and learned distribution via MMD or something similar
- There should be some idea of how the method compares against competitors, e.g. a table or figure showing the difference

[1] Song, Yang, et al. "Score-Based Generative Modeling through Stochastic Differential Equations." International Conference on Learning Representations. 2020.

**Questions:**

- Why is $z$ transposed in eq (6)?
- Can you change the $x’$ and $t’$ notation to $x_1$ and $t_1$ or similar? This would be more in line with other works that solve ODEs or SDEs
- Can you avoid using $\mathcal{N}$ to denote a network? This is usually used for normal distributions
- What is MAPE?
- Are you missing a minus in the exp in eq (31)?
- Am I right that the toy example of 5.1 involves no learning? Maybe make this clear
- What are $\phi^f$ and $\phi^b$ in algorithm 3? Please define them
- In line 3 of algorithm 3, should it be $\phi^b$ instead of $\phi$?
- Why do you need the discrete normalizing flow for the steady-state problem? Can’t you just start with a standard normal for $x_0$ and evolve it to the steady state? Do you know anything about what $x_0$ is learned by the flow (in theory and in practice)? What does this distribution look like in your experiments?
- What do you mean by “In future work, we may explore the application of PINF with diffusion for density estimation tasks to enhance model generalization and data generation quality.” in the conclusion? Is your aim to enhance diffusion generative models? How would your method help?

---

### Official Review · Reviewer_QfEf · 2023-10-30

**Soundness:** 2 fair
**Presentation:** 3 good
**Contribution:** 1 poor
**Rating:** 3
**Confidence:** 3

**Summary:**

The paper addresses the challenge of the normalization constraint in the Fokker-Planck (FP) equation, a vital equation describing the probability density function's (PDF) evolution in stochastic systems. Recognizing the potential of flow-based generative models, the authors introduce Physics-Informed Normalizing Flows (PINF), a novel extension of continuous normalizing flows (CNF). PINF integrates diffusion using the method of characteristics, reformulating the FP equation as ordinary differential equations (ODEs) to improve training efficiency and stability.

**Strengths:**

1.	PINF addresses the normalization constraint effectively, a challenge often faced in high-dimensional FP problems

2.	The model shows effectiveness over a few questions including high-dimensional cases.

**Weaknesses:**

1.	The problem formulation is not clear. There are 3 problem setups w./w.o. time varying/diffusion term. The drifting term seems to be a known function rather than to be learned. So what’s the point of the 1st setup (algorithm 1) here solving an PDE with explicit knowledge of time derivative equation?

2.	The paper motivation is not clear. As the data is generated by solving the ODEs, what are the benefits to use neural network to replace PDE solvers? Whether NN is faster/more robust/generalizable compared to direct solvers, it should be explicitly mentioned in the introduction and experimental comparisons should be provided.

3.	Physics-constraint and probability density conservation are mentioned multiple times in the paper and the author seems to claim it is ensured by the design of the NN formulation. However, I didn’t see any description or proof on this, along with experimental verification. Please correct me if I’m wrong.

4.	Lack of comparison baselines. There is no comparison baseline in the experiments, making it difficult to evaluate the difficulty of the task. Besides, there is no reproducibility check.

5.	The paper claims itself as “self-supervised training method” (under equation 19). This is questionable as the paper just uses ODE solver to generate labeled data during training, so it is essentially supervised training.

**Questions:**

1.	Under equation 19, “This structured design can impose hard constraints that are strictly enforced,”. can you describe which hard constraints are imposed? The initial condition match or the probability normalization constraint? Please elaborate if the latter.

2.	Under equation 22, “the total density integral on the space domain remains conserved as the solution evolves under equation constraints or ODEs controls”. Please elaborate, specifically what constraints the equations enforce.

3.	In figure 2, what is the difference between ground truth and ODE solution? Just integral error?

---

### Official Review · Reviewer_oz5q · 2023-10-31

**Soundness:** 2 fair
**Presentation:** 3 good
**Contribution:** 2 fair
**Rating:** 3
**Confidence:** 3

**Summary:**

The authors introduce a physics informed normalizing flow model for solving Fokker-Planck equations by using normalizing flows to solve the normalizing constraint.

**Strengths:**

The paper is well grounded theoretically in a natural extension to CNFs which are applied specifically to the TFP and SFP problems.

**Weaknesses:**

Though the results are interesting, there seems to be a lack of any sort of baseline compared to say standard PINNs (which the authors mention as falling after d=3). Most of the results are evaluated qualitatively where the authors observe good agreements between the true solution and the nn solution. A way to perhaps solidify results could be to analyze some sort of loss metric (perhaps average MAPE or max MAPE) for different models (PINF, PINN) as d is increased, and see the differences in drop off.

Also, some exploration for comparisons as to how p_net does as compared to p_ode on the TFP problems might help better understand the performance of the model.

Finally, the authors deal primarily with solutions for the Fokker Planck equation, which may be somewhat limited in scope in terms of applicability to the general conference.

**Questions:**

Why is only p_net returned for SFP as compared to both p_ode and p_net for TFP?

---

> ### Author Response · Authors · 2023-11-18
>
> Thanks a lot for your precious feedback.
>
> - Concerning the comparison with baselines, we have successfully reproduced both standard PINNs and PINNs incorporating hard constraints on initial conditions. We believe that enforcing the correct initial conditions can accelerate the accurate learning of the solution in TFP equations. Our experimental results indeed suggest that the PINN with hard constraints outperforms the standard PINN, yet challenges persist in higher dimensions, as illustrated in the accompanying figures.
> > [standard-PINNs.jpg](https://postimg.cc/G4F46PFF)
> > [PINNs-with-hard-constraints.jpg](https://postimg.cc/gXDqdF3K)
>
> This forms the foundation for our assertion that "For this problem, the PINN is effective primarily with the dimension d ≤ 3."
>
> - For your question, we understand our prediction modes need to be better clarified.
>
>   * Since the initial conditions of the TFP equation are given, employing precise initial values in conjunction with the forward evolution of ODEs yields an alternative prediction, denoted as $p_\text{ode}$. This reference prediction is often more accurate than $p_\text{net}$, albeit with higher computational costs.
>
>   * In SFP equations, where initial conditions are unavailable, it is unnecessary to employ an ODE solver to obtain $p_\text{ode}$, as initial values for ODEs are also predicted by the neural network instead of the ground truth.
>
> We genuinely appreciate your valuable suggestions, which are really helpful for our work.

---

### Meta-Review · Area_Chair_QTkb · 2023-12-06

**Metareview:**

This paper introduces Physics-Informed Normalizing Flows (PINF) as a novel extension of continuous normalizing flows to address the normalization challenge in solving the Fokker-Planck equation. By incorporating diffusion through the method of characteristics, PINF efficiently solves high-dimensional time-dependent and steady-state Fokker-Planck equations. The method is mesh-free and causality-free, offering a versatile solution for complex data distribution learning. While the proposed idea is interesting, there are lots of concerns on the presentation and technical quality of the paper and these concerns are unfortunately not well addressed in the rebuttal. I would suggest revising the manuscript according to the review comments and submit in future venues.

**Justification For Why Not Higher Score:**

There are lots of concerns on the presentation and technical quality of the paper

**Justification For Why Not Lower Score:**

N/A

---

### Decision · Program_Chairs · 2024-01-16

Reject